# Fusion Method and Application of Several Source Vibration Fault Signal Spatio-Temporal Multi-Correlation

Longhuan Cheng, Jiantao Lu *[ID], Shunming Li, Rui Ding, Kun Xu [ID] and Xianglian Li

College of Energy and Power Engineering, Nanjing University of Aeronautics and Astronautics, Nanjing 210000, China; 13121583471@163.com (L.C.); smli@nuaa.edu.cn (S.L.); Jerry.DingRui@gmail.com (R.D.); kunxu.nuaa@gmail.com (K.X.); lxle@nuaa.edu.cn (X.L.)

* Correspondence: lujt@nuaa.edu.cn

**Abstract:** Combined with other signal processing methods, related algorithms are widely used in the diagnosis and identification of rotor faults. In order to solve the problem that the vibration signal of a single sensor is too single, a new multi-source vibration signal fusion method is proposed. This method explores the correlation between vibration sensors at different locations by using multiple cross-correlations of spatial locations. First, wavelet noise reduction and linear normalization are used to process the original data. Then, the signal energy correlation function between the sensors is established, and the adaptive weight is obtained. Finally, the data fusion result is obtained. Taking rotor bearing and gear failures at different speeds as an example, the data of three vibration sensors at different positions are fused using the spatio-temporal multiple correlation fusion method (STMF). Through the intelligent fault diagnosis method stacked auto encoder (SAE), compared with single sensor data, average weighted fusion data and neural network fusion data, STMF method can reach a diagnosis accuracy of more than 94% at 700 rpm, 900 rpm and 1100 rpm. It is concluded that the result of the STMF method is more effective and superior.

**Keywords:** vibration signal; multiple cross-correlation; signal fusion; fault diagnosis

## 1. Introduction

With the development of the information age, the reliability requirements of mechanical equipment are getting higher and higher [1]. Rotating parts of mechanical equipment, such as shafts, gears and bearings, are prone to shaft cracks, gear breaks and bearing wear and other dangerous hazards [2]. In order to ensure the safe and reliable operation of the equipment, it is particularly important to obtain fault sources and fault data in time, and perform an accurate and effective fault diagnosis.

Signal correlation function was widely used in rotor system fault diagnosis—many scholars have done research in this area. A diesel engine fault warning method based on the correlation analysis of the envelope of cylinder head vibration signal was proposed [3–5], and a correlation analysis was proven to be able to better reflect the fault state of the unit. A maximum correlation kurtosis deconvolution algorithm (MCKD) was used to process generator stator vibration signals, and the extraction effect of fault signal characteristic frequency was improved [6]. S. Janjarasjitt et al. [7] proposed a partial correlation integral algorithm, and used the size index calculated by the algorithm to successfully predict the faults of rolling bearings in rotating machinery. M. Rahmatian et al. [8] used the obtained ground current signal for feature selection based on cross-correlation technology. Then, they used these functions to perform fault detection on the four-layer multiplier perceptron artificial neural network (ANN) training, and obtain a good diagnosis. Hong et al. [9] proposed a method for gearbox fault feature extraction based on empirical mode decomposition (EMD) and multi-fractal detrended cross-correlation analysis (MFDCCA), which can successfully achieve better fault diagnosis accuracy.

The fault information obtained by a single sensor is single, while multiple different sensors can obtain fault information from multiple angles. Sensor data from multiple locations or different angles are also used to extract unsupervised features in the frequency domain and detect engine failures in industrial environments [10,11]. In addition, multi-sensor systems can simultaneously detect sensors or deal with faults through Bayesian networks, and quantify the reliability of sensor data through network weights [12,13]. Hua et al. [14] proposed a conversion method that converts vibration signals from multiple sensors into images. Using experimental data compared with other methods, they found that this method has higher recognition accuracy and faster convergence speed. For gearbox failures of multi-source signals, Zhi et al. [15] proposed a new tensor classifier, called the core flexible displaceable convex hull-based tensor (KFDCH-TM), which used wavelet packet transform (WPT) from multiple sources. The feature tensor extracted from the source signal is used to diagnose gearbox faults through KFDCH-TM.

With the development of fault diagnosis and signal processing technology, signal fusion technology is constantly updated and optimized. In the traditional multimodal sensor fusion method, data fusion can be achieved by manually extracting the features of multimodal sensors and simply connecting the growth vectors [16–18]. Research on multi-mode sensor fusion using deep learning networks for fault diagnosis has only begun to appear in recent years. S.Ma et al. [19] proposed a deep-coupled auto encoder network for vibration and acoustic sensor data fusion to realize the fault diagnosis of gears and bearings. S.Ma et al. [20] constructed a deep-coupling restricted Boltzmann machine for data fusion of vibration and acoustic emission sensors in order to diagnose tool faults. S. Hao et al. [21] proposed a multi-sensor fusion algorithm using the CNN-LSTM network to monitor bearing faults. Hui et al. [22] used a DCNN-based two-way vibration signal data fusion method for the health status recognition of planetary gearboxes. Due to the possibility of misclassification, more vibration signal fusion needs to be considered. In addition, the sensor installation location to reduce the redundancy of fusion data was studied [23,24].

The proposed multi-vibration sensor signal fusion method can be used for different rotor faults. Based on the established multi-sensor spatial position correlation model and signal energy correlation model, the time deviation and adaptive weights of different sensors are estimated to achieve the different adaptive fusion of faults. The fault diagnosis method of SAE is used to classify the fused fault data. Compared with the data classification results of unfused single sensor data and other fusion methods, the results show the superiority of the fusion method.

The main innovations are as follows:

(1) A model of spatial correlation of multiple vibration sensors is proposed;
(2) Propose a method of spatial and temporal multi-correlation fusion of vibration sensors;
(3) The proposed method can handle different types of rotor faults and can effectively implement fault diagnosis.

The organization of the rest of this article is as follows: The second part introduces the basic wavelet noise reduction theory, linear normalization method and Hilbert–Huang transform. The fourth part verifies the STMF method and the comparative analysis with other fusion methods, which verifies the superiority of the STMF method. The fifth part is the conclusion. The overall framework and ideas of the content in the article are shown in Figure 1. First, wavelet denoising was performed on the original fault data and a linear normalization method was used to standardize the data. Then, the data are fused and the weight is determined by the correlation of the multi-sensor spatial position coordinates and the energy correlation of each sensor. Finally, the SAE method was used to intelligently diagnose faults with the fusion fault data.

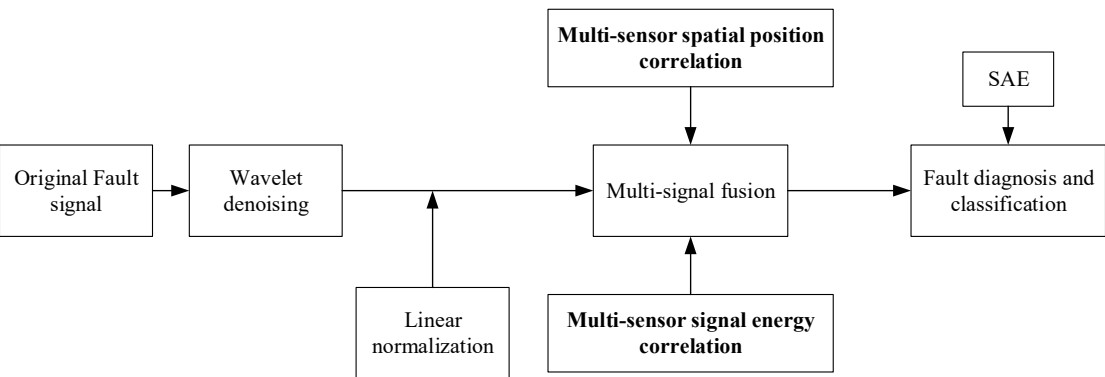

**Figure 1.** The overall framework and ideas of the content.

## 2. Introduction to Basic Theory

### 2.1. Wavelet Noise Reduction

Using wavelet change threshold method to reduce raw signal noise, its advantage is that the noise can be basically suppressed, and the characteristic information of the original signal is better preserved.

Wavelet transform is to decompose a signal into a series of superposition of wavelet functions (or fitting of wavelet functions of different scales and times) [25]. These wavelet functions are all obtained by a mother wavelet after translation and scale expansion [26–29]. It takes a function called mother wavelet as the position $\tau$, and the signal was measured $x(t)$ as the inner product under different scales $\alpha$, namely:

$$WT_x(\alpha, \tau) = \frac{1}{\sqrt{\alpha}} \int_{-\infty}^{+\infty} x(t)\varphi\left(\frac{t-\tau}{\sigma}\right)dt \tag{1}$$

where $\alpha$ is called the scale factor, and $\alpha > 0$, which is to stretch the basic wavelet function. While $\tau$ reflects the displacement, whose value can be positive or negative. Both $\alpha$ and $\tau$ are continuous variables. The frequency domain corresponding to Equation (1) is expressed as:

$$WT_x(\alpha, \tau) = \frac{\sqrt{\alpha}}{2\pi} \int_{-\infty}^{+\infty} x(\omega)\psi(\alpha\omega)e^{+j\omega\tau}d\omega \tag{2}$$

The obtained original single fault signal is obtained by wavelet transformation to obtain $\omega_{j,k}$, then, $\hat{\omega}_{j,k}$ is obtained through the process of threshold transformation, and finally, the noise-reduced fault signal is obtained by the method of wavelet reconstruction. The process of wavelet threshold noise reduction is shown in Figure 2.

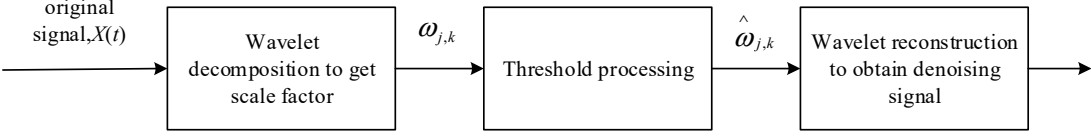

**Figure 2.** The process of wavelet threshold noise reduction.

### 2.2. Linear Normalization

Many dimensionality removal methods were listed in the literature [30], including standardization method, extreme value method, linear ratio method, normalization method, vector normalization method and power coefficient method. Each method has its advantages and disadvantages. This paper combines the extremum processing method

in the literature [31] to reduce dimensionality the fusion signal. Linear transformation is performed on the fused data to map it to [0,1]. The results of normalization are as follows:

$$X' = \frac{X - \min_X}{\max_X - \min_X}$$

(3)

where $X'$ is the normalized result, $\min_X$ and $\max_X$ are the minimum and maximum values in the sample. The data after normalization does not have dimensions, but the data of each sensor is mapped to [0,1], which will not change the trend and regularity between the data before normalization.

### 2.3. Hilbert–Huang Transform

The Hilbert–Huang transform (HHT) is used to draw the envelope spectrum of the test vibration signal, which can more easily view the fault characteristics of each component.

The main content of Hilbert–Huang transformation consists of two parts. The first part is empirical mode decomposition (EMD), which was proposed by Huang; the second part is Hilbert spectrum analysis (HSA). The HHT of the signal $x(t)$ is as follows:

$$\overset{\wedge}{x} = \frac{1}{\pi} \int\limits_{-\infty}^{+\infty} \frac{x(\tau)}{t - \tau} d\tau = \frac{1}{\pi} \int\limits_{-\infty}^{+\infty} \frac{x(t - \tau)}{\tau} d\tau = x(t) \times \frac{1}{\pi t}$$

(4)

The HHT is equivalent to an interesting filter, in which the amplitude of the spectral components remains unchanged, but their phases are shifted by one [32,33]. Its unit impulse response is $1/\pi t$.

The inverse of the HHT is as follows:

$$x(t) = -\overset{\wedge}{x}(t) \times \frac{1}{\pi t}$$

(5)

The HHT has the following properties:

(1) The signal $x(t)$ and its HHT energy spectrum and amplitude spectrum are the same, but the phase spectrum is different.

(2) Signal $x(t)$ and $\overset{\wedge}{x}(t)$ are orthogonal to each other, namely:

$$\int\limits_{-\infty}^{+\infty} x(t)\overset{\wedge}{x}(t)dt = 0$$

(6)

(3) If you perform $x(t)$ twice using HHT, you will get $-x(t)$, namely $\overset{\wedge}{\overset{\wedge}{x}}(t) = x(t)$.

## 3. Multi-Sensors Spatio-Temporal Multi-Correlation Fusion Analysis

### 3.1. Spatial Correlation Analysis

Literature [34] pointed out that when measuring the same signal source, different sensors would affect the consistency of space and time, which resulted in time delay and direction error. Elimination of time deviation can ensure the synchronization of data measured by heterogeneous sensors. Figure 3 shows a schematic diagram of measuring the same signal source with heterogeneous sensors.

Where $S(t)$ is the measured signal source, $X(t)$, $Y(t)$, $Z(t)$ are the signals measured, respectively by the three sensors. It can be seen from the figure that the propagation speed is different and there is a time deviation due to the different distances between the sensors and the same signal source. In order to eliminate the influence of time deviation, it is necessary to study the spatial correlation between sensors. The space coordinate system was established with the signal source as the origin, and three sensor spatial coordinates

were obtained. The relationship between the signal source and the position parameters of the three sensors is shown in Table 1.

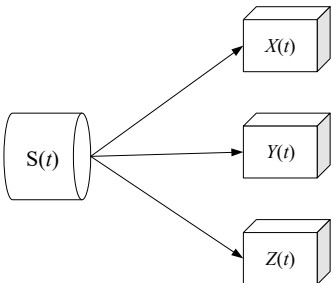

**Figure 3.** Different types of sensors measure the same signal source.

**Table 1.** Spatial location parameters of three different sensors.

|            | Sensor 1 | Sensor 2 | Sensor 3 |
|------------|----------|----------|----------|
| X distance | 0        | $a_2$    | $a_3$    |
| Y distance | $b_1$    | 0        | $b_3$    |
| Z distance | $c_1$    | $c_2$    | $c_3$    |

The established signal source and sensor position distribution diagram is shown in Figure 4. Assume that there is only translation, not rotation, between the three sensors. In the space coordinate system, the position coordinates of the sensors are known. The direction distance of X, Y, Z can be regarded as the X coordinate, Y coordinate and Z coordinate of the three sensors. The spatial coordinates of three sensors are established to find out the spatial position correlation.

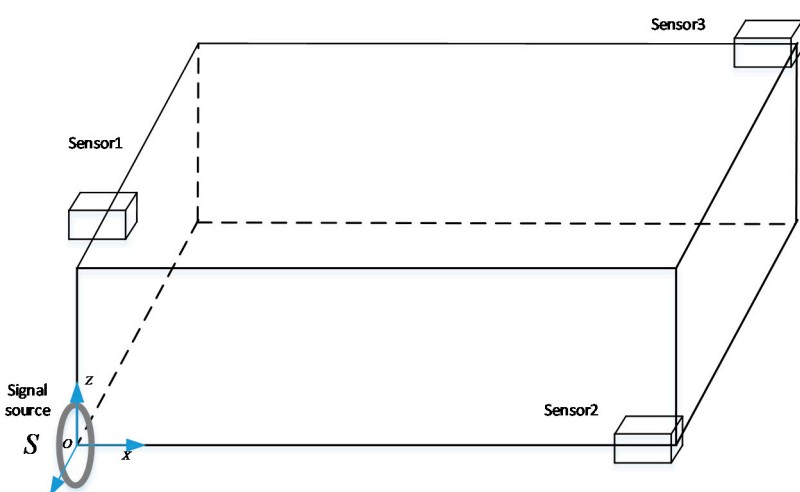

**Figure 4.** Position distribution of signal sources and sensors.

The spatial coordinate matrix of the three sensors is:

$$
\begin{bmatrix}
X_1 & X_2 & X_3 \\
Y_1 & Y_2 & Y_3 \\
Z_1 & Z_2 & Z_3
\end{bmatrix}
=
\begin{bmatrix}
0 & a_2 & a_3 \\
b_1 & 0 & b_3 \\
c_1 & c_2 & c_3
\end{bmatrix}
\tag{7}
$$

The coordinate transformation relationship between sensor 1 and sensor 2 and between sensor 1 and sensor 3. The position distribution of signal sources and sensors is shown in Figure 4.

$$
\begin{bmatrix} X_1 \\ Y_1 \\ Z_1 \end{bmatrix} = \begin{bmatrix} X_2 \\ Y_2 \\ Z_2 \end{bmatrix} + \begin{bmatrix} -a_2 \\ b_1 \\ c_1 - c_2 \end{bmatrix}, \quad \begin{bmatrix} X_1 \\ Y_1 \\ Z_1 \end{bmatrix} = \begin{bmatrix} X_3 \\ Y_3 \\ Z_3 \end{bmatrix} + \begin{bmatrix} -a_3 \\ b_1 - b_3 \\ c_1 - c_3 \end{bmatrix} \tag{8}
$$

Equation (8) is used to unify the spatial information of the three sensors into the coordinate system of sensor 1, which can eliminate the time deviation between different sensors.

### 3.2. Time Deviation Correlation

Multi-sensor data fusion requires timely synchronization. According to the above spatial coordinate relation, the influence of spatial position between the two sensors is eliminated and the data synchronization is realized. Spatial position correlation is used to estimate the time deviation between two sensors [35]. Taking sensor 1 as reference, the time deviation between sensor 1 and sensor 2, and the time deviation between sensor 1 and sensor 3 are,= respectively expressed as follows:

$$
\tau_1 = \frac{1}{3}\left( \frac{X_2 - X_1}{X_1} + \frac{Y_2 - Y_1}{Y_1} + \frac{Z_2 - Z_1}{Z_1} \right), \quad \tau_2 = \frac{1}{3}\left( \frac{X_3 - X_1}{X_1} + \frac{Y_3 - Y_1}{Y_1} + \frac{Z_3 - Z_1}{Z_1} \right) \tag{9}
$$

In the above equations, $\tau_1$ and $\tau_2$ were, respectively, the time deviation between sensor 1 and sensor 2, and the time deviation between sensor 1 and sensor 3. Substitute Equation (8) into Equation (9) to obtain:

$$
\tau_1 = \frac{1}{3}\left( \frac{a_2}{X_1} + \frac{-b_1}{Y_1} + \frac{c_2 - c_1}{Z_1} \right), \quad \tau_2 = \frac{1}{3}\left( \frac{a_3}{X_1} + \frac{b_3 - b_1}{Y_1} + \frac{c_3 - c_1}{Z_1} \right) \tag{10}
$$

### 3.3. Temporal Correlation Analysis

The above formula is used to combine the signal with the time deviation to obtain the synchronization signals $x(t)$, $y(t + \tau_1)$ and $z(t + \tau_2)$ of the three sensors. $x(t)$, $y(t + \tau_1)$ and $z(t + \tau_2)$ signals are deterministic signals with limited energy, so the correlation coefficient between $x(t)$ and $y(t + \tau_1)$, $x(t)$ and $z(t + \tau_2)$ is defined as:

$$
d_{xy} = \frac{\sum\limits_{n=0}^{\infty} x(t)y(t + \tau_1)}{\left[ \sum\limits_{n=0}^{\infty} x^2(t) \sum\limits_{n=0}^{\infty} y^2(t + \tau_1) \right]} = \frac{R_{xy}}{\sqrt{E_x E_y}} \tag{11}
$$

$$
d_{xz} = \frac{\sum\limits_{n=0}^{\infty} x(t)z(t + \tau_2)}{\left[ \sum\limits_{n=0}^{\infty} x^2(t) \sum\limits_{n=0}^{\infty} z^2(t + \tau_2) \right]} = \frac{R_{xz}}{\sqrt{E_x E_z}} \tag{12}
$$

where the denominator $\sqrt{E_x E_y}$ was the square root of the normalized energy product of the signals $x(t)$ and $y(t + \tau_1)$, and the denominator $\sqrt{E_x E_y}$ was the square root of the normalized energy product of the signals $x(t)$ and $z(t + \tau_2)$. It can be seen from the formula that the power of the denominator energy is a constant, so the size of the correlation coefficient is determined by the numerator. $R_{xy}$ is correlation function of the signal $x(t)$ and $y(t + \tau_1)$. It can be obtained $|d_{xy}| \le 1$ according to Schwartz's inequality, when the two signals are completely equal, the numerator is equal to 1, $d_{xy} = 1$; when the two signals are not completely equal, $d_{xy} < 1$; when the two signals are completely unrelated $d_{xy} = 0$. In other words, $d_{xy}$ can be used to describe the degree of similarity between two signals.

The correlation function of vibration signals describes the correlation of two sample functions at different moments, which reflects the close relationship between two ran-

dom vibration signal waveforms when they move on the time coordinate [36]. The cross correlation function of discrete random vibration signals is expressed as:

$$R_{xy}(t) = \frac{1}{N-m} \sum_{t=1}^{N-m} x(t)y(t+m)(m=1,2,\cdots,k) \tag{13}$$

where $N$ is the number of sampling points, and the size of the cross-correlation function directly reflected the correlation between the two signals and it was a measure of waveform similarity.

### 3.4. Signal Fusion Based on Spatio-Temporal Multiple Cross-Correlation and Correlation Function

Multiple sensors with the same accuracy are used to measure different parts of the same object, and the effectiveness of the information collected may be inconsistent. It is obviously inappropriate to distribute the weight of the sensor according to a precision characteristic when an unexpected situation occurs in it, such as a large difference between the collected data and the actual situation, or damage leading to a complete failure. Since the correlation function of vibration signals is a measure of waveform similarity, it can also be seen as the support of one signal to another. Therefore, this support is used as the basis for weight assignment, that is, data fusion based on the starting point of the correlation function weighting method [37]. The implementation steps of the method in this section are shown in Figure 5.

In the fusion of traditional weighting algorithms, the most difficult thing is to determine the weight. Generally, the weighted average method is used to directly fuse the data of various sensors, which is not strict enough. On the contrary, the weighted value adjustment based on multiple cross-correlation functions is more accurate.

In this method, the accuracy of the sensor and the influence of random factors are taken into account to connect any signal with other signals. The greater the correlation, the greater the weight assigned, and vice versa. In general, the more accurate the information reflecting the target state, the higher the degree of support. The energy of the correlation signal is used to represent the correlation degree; that is, the higher the energy, the higher the correlation degree. The energy of discrete signals $x(t)$ and $y(t+\tau_1)$ is calculated by the following formula:

$$E_{xy} = \sum_{i=1}^{n} \left[ R_{xy}(i) \right]^2 \tag{14}$$

Assuming that the energy of the signal obtained by performing the cross-correlation operation of each signal is $E_{ij}$, the total correlation energy of the signal collected by the sensor $i$ and the signal collected by other sensors can be expressed as:

$$E_i = \sum_{j=1,j\neq i}^{n} E_{ij} \tag{15}$$

When the weight $p_i$ is proportional to the energy of the correlation function, there were:

$$\begin{aligned} p_1 : p_2 : \cdots : p_n = E_1 : E_2 : \cdots : E_n \\ p_1 + p_2 + \cdots + p_n = 1 \end{aligned} \tag{16}$$

Signals $x_1, x_2, \ldots, x_i$ from the same fault source acquired by different sensors, the different weights of each signal can be obtained from the above formula, so the final fusion result $X$ can be obtained as:

$$X = p_1 x_1 + p_2 x_2 + \cdots + p_i x_i \tag{17}$$

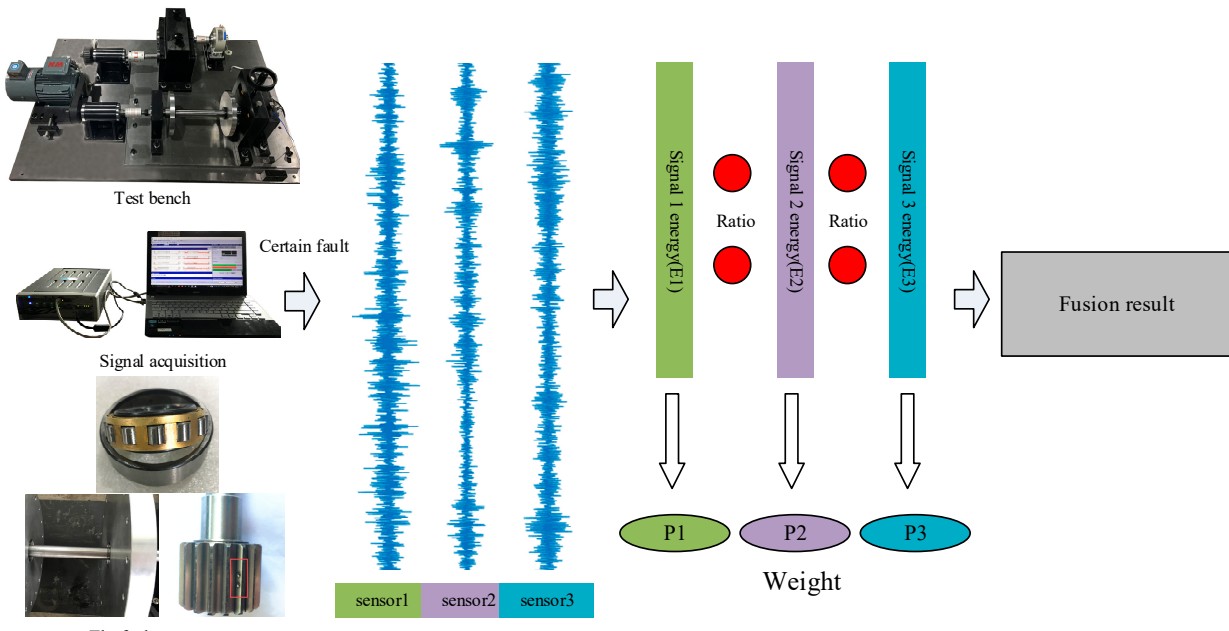

**Figure 5.** Steps of the fusion method.

## 4. Experimental Verification

The fault data obtained by a single sensor are not comprehensive. Multiple different sensors can obtain fault information from different angles, which is more conducive to enhancing the characteristics of the fault and improving the effect of fault diagnosis.

The time offset obtained by correlating the spatial positions of multiple sensors with the energy of multiple sensors is correlated to obtain the fusion data. In order to verify the effectiveness of the fusion method, SAE [38] deep learning method in the vibration test bench was used to compare and verify the original single data (OSD), average weighted fusion (AWF), neural network fusion (NNF) and STMF methods, and the superiority of STMF method is highlighted.

### 4.1. Experimental Device and Fault Setting

The test equipment and instruments are all placed in the open workshop, so the noise is very close to the actual situation, and the obtained fault data is an approximate simulation of the actual fault. In the signals of multiple sensors, the time deviation is eliminated, the linear normalization is processed, the correlation function is used to obtain the adaptive weight, and the test data are used for verification. One motor, two three-direction acceleration sensors (sensor 1 and sensor 2) and one unidirectional acceleration sensor (sensor 3) are used in the experiment. The sampling frequency in the experiment was 25.6 Khz. As shown in Figure 6, Figure 6a is the composition of the test bench and Figure 6b is the position arrangement and coordinate axis of the three sensors. The gearbox contains two gears: planetary and solar. Specific gear parameters are shown in Table 2. The speed of the drive motor is 0~1500 rpm, the transmission ratio of the reducer is 3, and the output speed of the reducer is 0~4500 rpm. HRB6206 deep groove ball bearing specific parameters are shown in Table 3.

The types of bearing faults are shown in Figure 7, including 0.6 mm cracks in the inner ring and 0.6 mm cracks in the outer ring of the bearing. The gear fault types are shown in Figure 8, including planetary gear crack fault (1.2 mm, 3.6 mm), planetary gear pitting fault (3.6 mm), and planetary gear wear fault (1.2 mm, 3.6 mm). Taking the center position of the bottom of the bearing seat as the origin, the spatial coordinate system is established. The establishment of the space coordinate system of the test bench is shown in the red line part of Figure 6b. Sensor 1 and sensor 2 are respectively placed on the left and right bearing

seats in Figure 6a, as shown in Figure 6b for specific positions. The coordinates are [0,0,20] and [40,0,20]. Sensor 3 is placed at the spatial coordinates [15,0,0] in centimeters. Unit of measurement: cm.

Various fault data obtained through experiments can be used to obtain time domain graphs of different faults through signal processing methods, but only the time domain graphs cannot see the difference of different faults.

Therefore, the time domain signal is mapped to the frequency domain through the HHT, and then the envelope spectrum of different faults is obtained, which can clearly see the difference of different faults. According to the test data at 1300 rpm, the planetary gear crack fault (3.6 mm) and wear fault (3.6 mm), 0.6 mm cracks in the inner ring and 0.6 mm cracks in the outer ring of the bearing, and the time domain diagrams and frequency domain envelope spectra of the four faults are shown in Figure 9. Among them, the horizontal axes of the time domain sampling point graph reveal the number of sampling points, the horizontal axes of the frequency domain envelope spectrogram reveal frequency (Hz), and the longitudinal axes of both reveal the amplitude (mv).

According to the calculation formula of theoretical gear failure frequency and bearing failure frequency combined with the component layout of the test bench, the failure frequencies of these four types of failures can be obtained as: 21.67 Hz, 21.67 Hz, 53.4 Hz and 159.64 Hz. In Figure 9, the actual fault frequencies corresponding to the four types of faults are 18.36 Hz, 18.36 Hz, 55.08 Hz and 164.9 Hz. The theoretical fault frequency and the actual fault frequency are closely within the allowable range, so the data obtained by the test can be used normally.

### 4.2. Analysis of Test Results

The maximum speed that the test motor can reach is 1500 rpm, and different motor speeds are set in the test, which are 700 rpm, 900 rpm, 1100 rpm, 1300 rpm. Among them, 1300 rpm is the rotation speed closest to the resonance frequency of the motor.

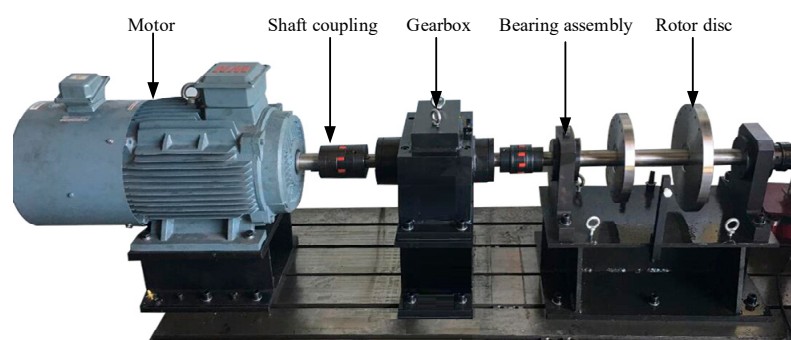
(**a**) Components of the test bench

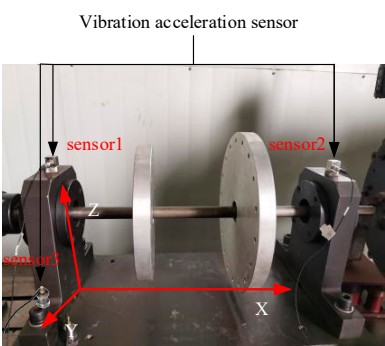
(**b**) Sensor mounting position

**Figure 6.** Test Bench.

**Table 2.** Parameters of gear.

| Gear | Number of Teeth | Modulus (mm) | Engagement Angle (deg.) | Material |
| --- | --- | --- | --- | --- |
| Sun gear | 55 | 2 | 20 | S45C |
| Planetary gear | 75 | 2 | 20 | S45C |

**Table 3.** The basic parameters of deep groove ball bearings.

| Parameters | Inner Ring Diameter | Outer Ring Diameter | Rolling Body Diameter | Pitch Diameter | Number of Rolling Elements |
| --- | --- | --- | --- | --- | --- |
| Value | 30 mm | 62 mm | 9.6 mm | 46 mm | 9 |

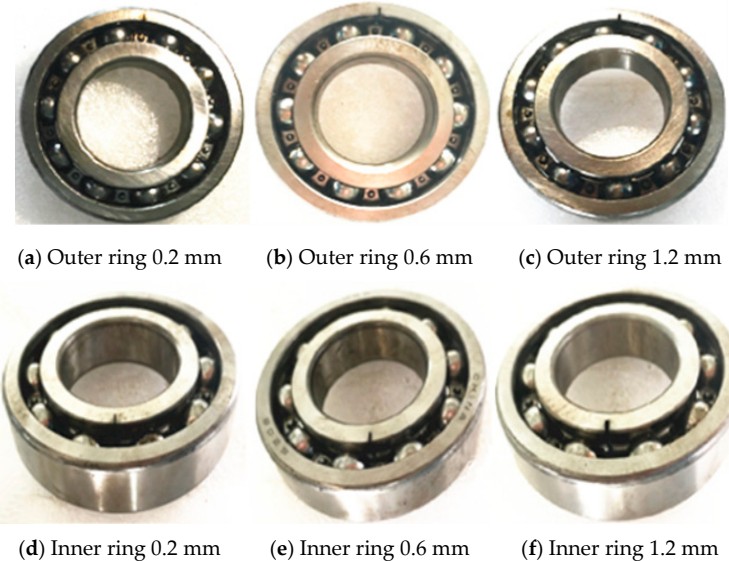

(**a**) Outer ring 0.2 mm　　(**b**) Outer ring 0.6 mm　　(**c**) Outer ring 1.2 mm

(**d**) Inner ring 0.2 mm　　(**e**) Inner ring 0.6 mm　　(**f**) Inner ring 1.2 mm

**Figure 7.** Bearing fault.

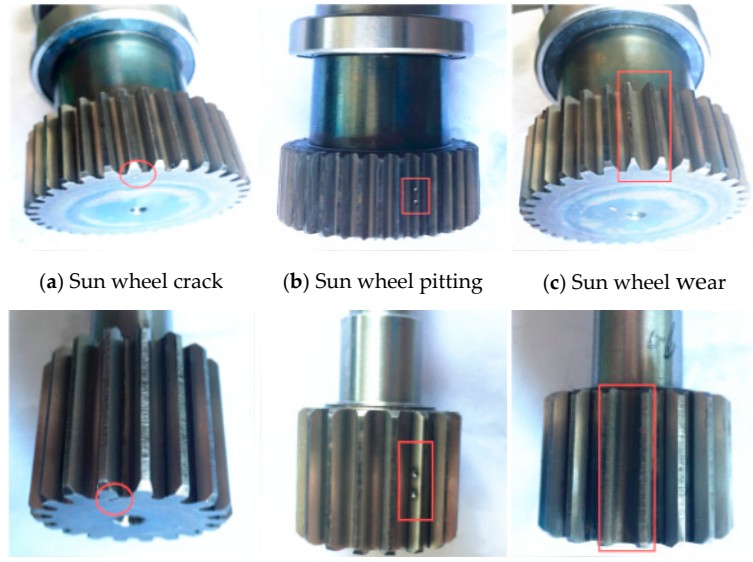

(**a**) Sun wheel crack　　(**b**) Sun wheel pitting　　(**c**) Sun wheel wear

(**d**) Planetary wheel crack (**e**) Planetary wheel pitting (**f**) Planetary wheel wear

**Figure 8.** Gear failure.

The test working conditions are set to observe whether the fault results will change differently here. The test condition of 1500 rpm is too dangerous, so there is no need to set it. The SAE fault diagnosis method is used to compare and verify the original data and fusion data of each working condition. In addition, the traditional AWF and NNF methods are compared with the STMF method, which highlights the superiority of STMF data.

When the fault data are verified, the fault type has normal data (*n*), bearing fault, planetary gear fault. The types of bearing faults are included 0.6 mm cracks in the inner ring (*nq0.6*) and 0.6 mm cracks in the outer ring of the bearing (*wq0.6*). The types of planetary gear fault include planetary gear crack 1.2 mm fault (*dlw*), planetary gear crack 3.6 mm fault (*xlw*), planetary gear pitting fault 3.6 mm (*6ds*), planetary gear wear 1.2 mm fault (*xms*), planetary gear wear 3.6 mm fault (*dms*). In Figures 10–13, the dimension1 of the horizontal coordinate refers to the fault feature 1 after dimensionality reduction, and the dimension 2 of the longitudinal coordinate refers to the fault feature 2 after dimensionality reduction.

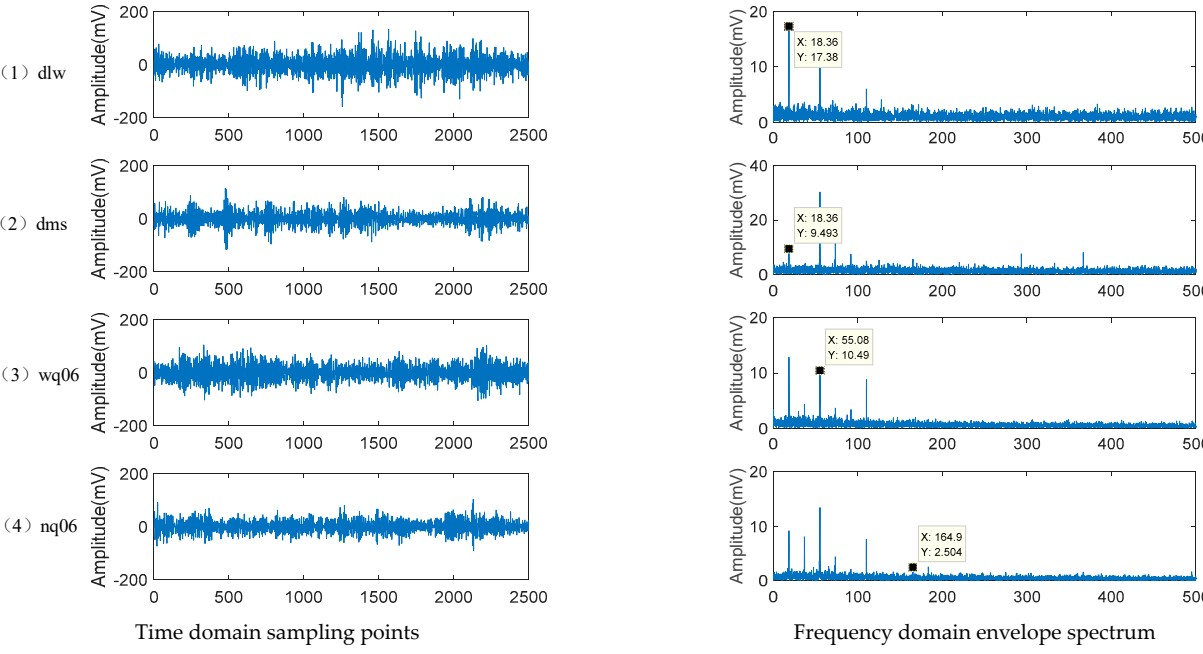

**Figure 9.** Time domain and envelope spectrum graphs with signals of different health conditions.

(1)    700 rpm test conditions

Under the working condition of 700 rpm, the classification results of the above eight types of faults using different methods are shown in Figure 10.

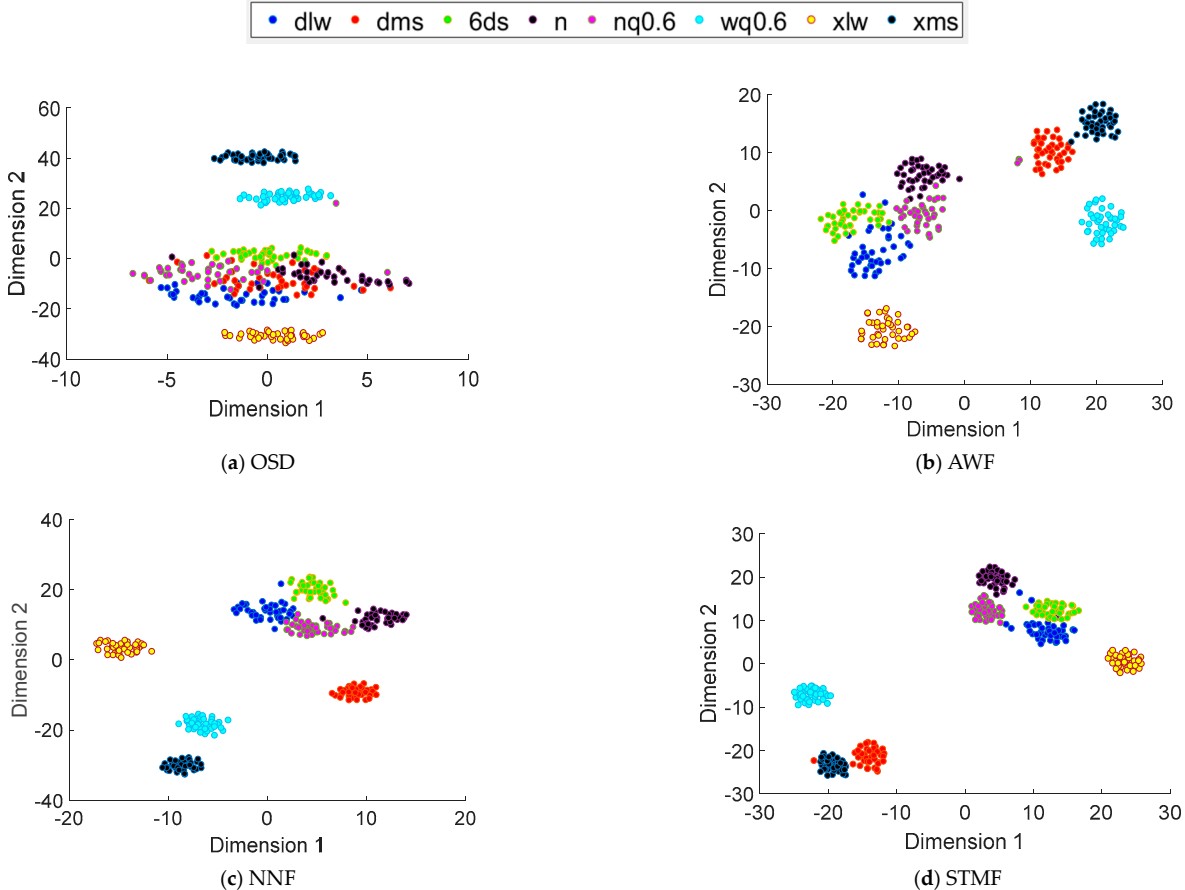

**Figure 10.** Comparison of eight types of fault classification results under 700 rpm.

The OSD, AWF, NNF and STMF data were verified by the fault diagnosis method of SAE. It can be seen that, when the motor speed is 700 rpm, only three types of fault in the original data of a single sensor are well separated in OSD, while the other five types of fault are all mixed together, and the classification effect is poor. In comparison, the classification effect of the data obtained by the AWF is better. However, the clusters of each type of failure do not aggregate well. Compared with the first two kinds of data, the data of NNF are easier to be classified and clustered.

The proposed STMF method can determine the weight in the case of direct correlation analysis of real-time data, without any prior knowledge of sensor measurement data and the influence of some external factors, which is easy to implement. The fault diagnosis classification graph obtained by the STMF method is not only good for each type of fault, but also for the compactness of each type of fault.

(2)    900 rpm test conditions

Under the working condition of 900 rpm, the classification results of the above eight types of faults using different methods are shown in Figure 11.

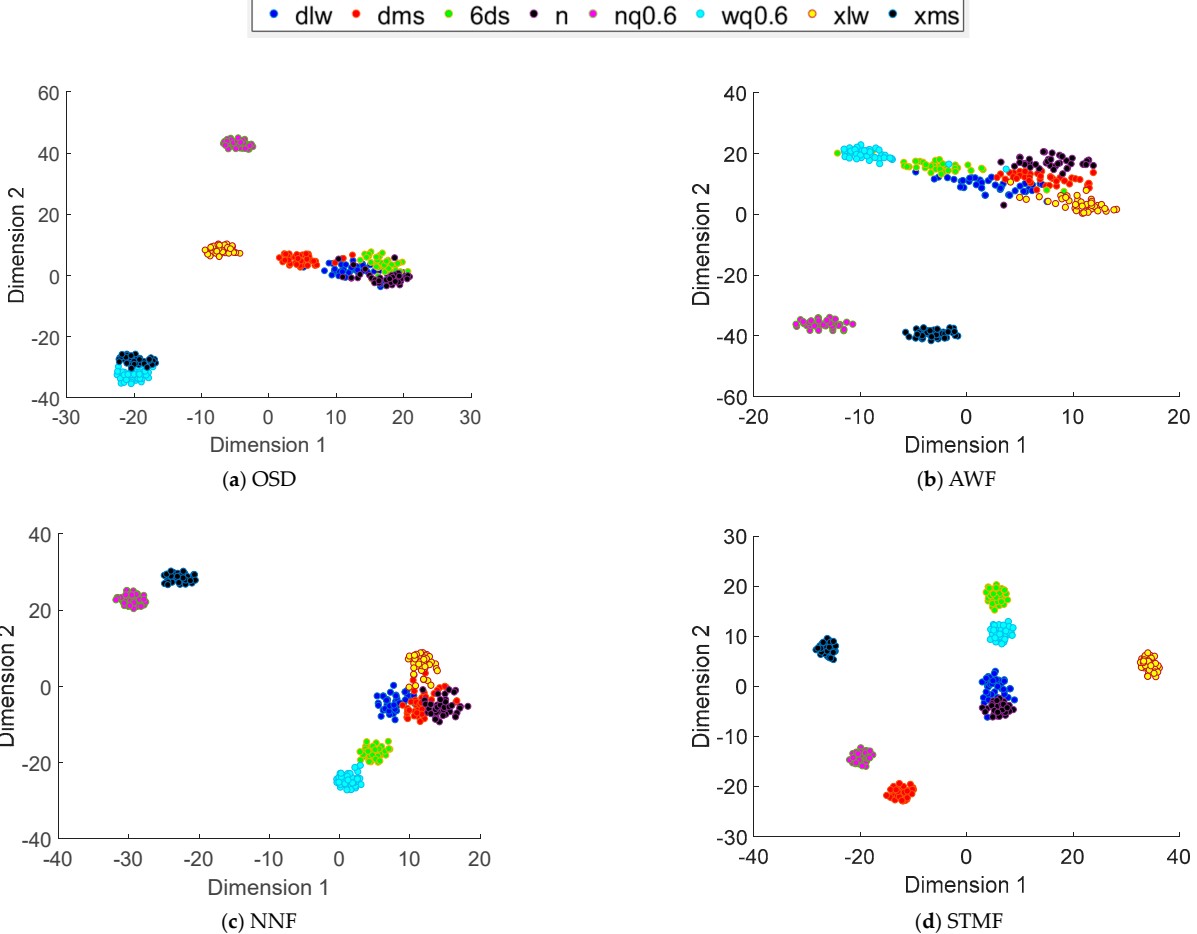

**Figure 11.** Comparison of eight types of fault classification results under 900 rpm.

The classification effect of OSD is average, and several types of faults are clustered together under the working condition of 900 rpm. The aggregation degree of each fault in the AWF method is relatively poor, and the classification results of several faults are not very good; only *xms* and *nq0.6* are separated from other faults. In NNF, the four types of fault classification results of planetary gear cracks (1.2 mm and 0.6 mm) and wear and bearing inner ring cracks of 0.6 mm are poor, all of which will affect the accuracy of the final fault diagnosis.

In STMF fault diagnosis, in addition to large cracks in the planetary gear and poor normal classification, other faults are well classified, and the aggregation effect of each fault is also excellent, which greatly improves the accuracy of fault diagnosis.

It can be seen from Figure 11 that under the working condition of 1100 rpm, the diagnosis effect of the original fault data is poor; only the faults *xms* and *wq0.6* are separated from other faults, and the other faults have poor diagnosis effect, which is inferior to the NNF method. The classification effect is similar. AWF has a better diagnostic effect for each type of fault, but the aggregation of individual faults is poor, which makes each type of fault appear discrete.

(3)    1100 rpm test conditions

Under the working condition of 1100 rpm, the classification results of the above eight types of faults using different methods are shown in Figure 12.

Based on the classification of the original data, the classification effect of the NNF method is better. The fault data classification effect obtained by the STMF method is the best. Not only can the various types of faults be separated, but the aggregation degree of each type of fault is also better, which reflects the strong dynamic adaptability of the STMF method and greatly improves the accuracy of fault diagnosis.

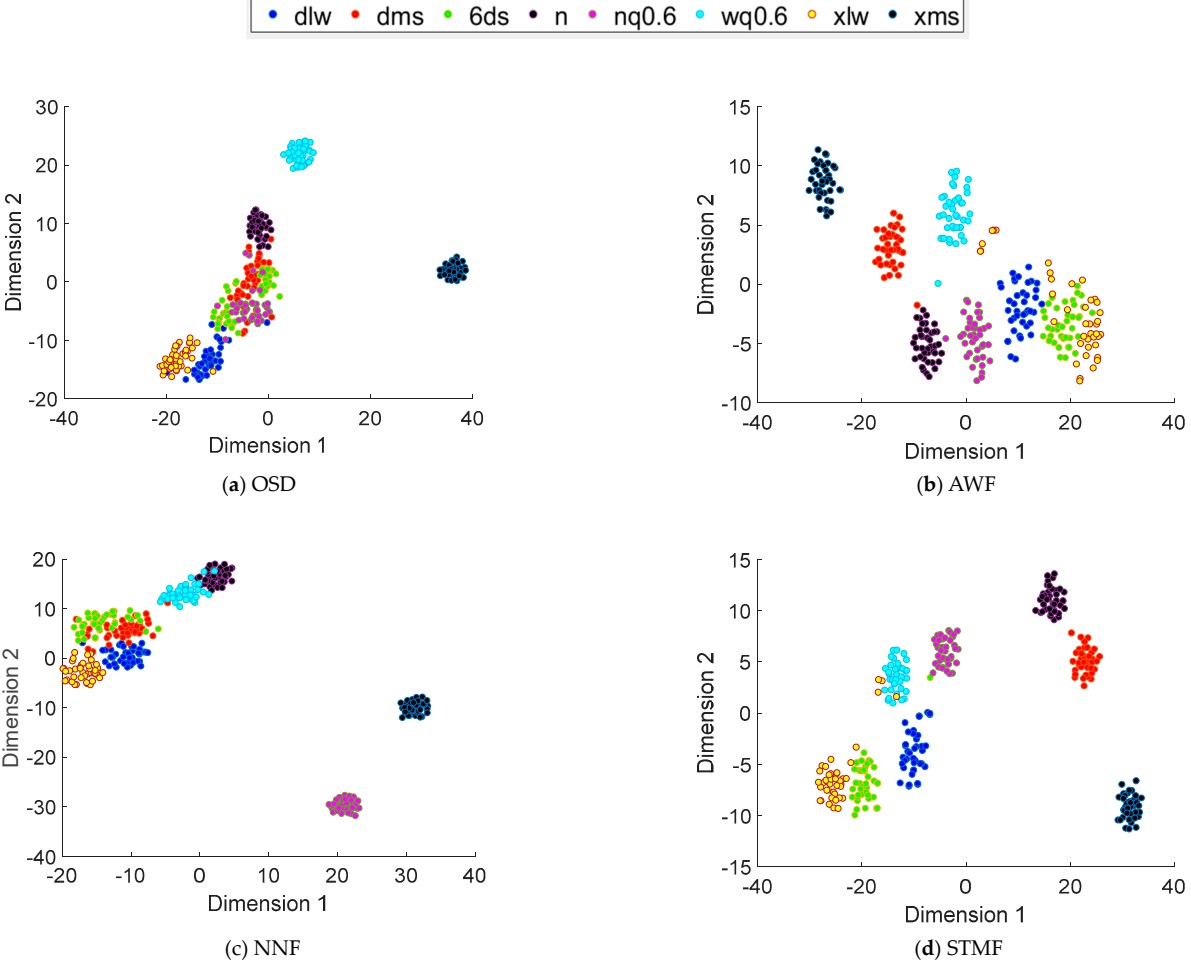

**Figure 12.** Comparison of eight types of fault classification results under 1100 rpm.

(4)    1300 rpm test conditions

Under the working condition of 1300 rpm, the classification results of the above eight types of faults using different methods are shown in Figure 13.

Under the working condition of 1300 rpm, in the original data, only *n* is separated, and the other fault data are divided into two piles and mixed together. In the fault diagnosis of AWF and NNF, only *nq0.6* and *xms* can be separated from other faults, and the rest are grouped together. In general, the fault diagnosis effects of OSD, AWF and NNF are not very satisfactory, resulting in several types of faults that cannot be separated.

In the effect diagram of STMF diagnosis, although each type of fault looks relatively discrete, all types of faults have been separated. In contrast, the fault diagnosis effect of OSD, AWF and NNF is better, but the aggregation effect of each type of fault needs to be improved.

In order to highlight the effectiveness of the STMF method, in addition to comparing the single sensor raw data, the traditional AWF data and the fault diagnosis classification graph of the NNF data, the fault diagnosis accuracy and the variance of the accuracy of each method are also counted. The establishment of the fault diagnosis accuracy and variance of each working condition and fusion method are shown in Table 4.

It can be seen from Table 4 that under different speed conditions, the data obtained by the STMF method has the highest accuracy in the SAE fault diagnosis, and the accuracy of the diagnosis is very stable, so the variance of the diagnosis accuracy is very small, which is represented by a thin pink bar in Figure 14.

In comparison, the diagnostic accuracy of OSD, AWF, and NNF data is lower than that of the STMF method. The accuracy of data diagnosis of the NNF method is higher than that of AWF and OSD, but the accuracy variance is the largest among several methods. Yes, the stability is poor. Figure 12 shows the test times and accuracy graphs of several other methods.

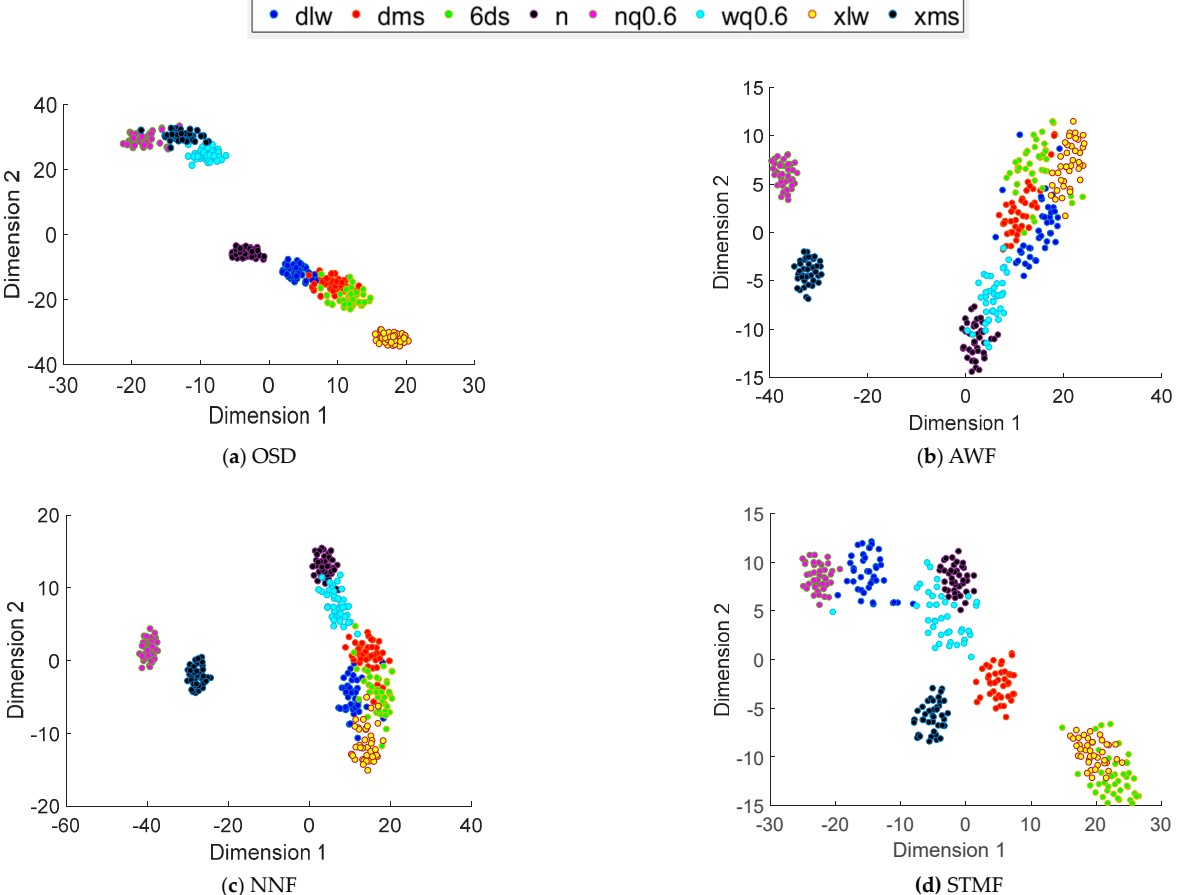

**Figure 13.** Comparison of eight types of fault classification results under 1300 rpm.

**Table 4.** SAE handles the fault classification accuracy rate of different data.

| Working Condition | Fusion Method | Total Sample | Training Samples | Testing Sample | Accuracy |
|---|---|---|---|---|---|
| 700 rpm | OSD<br>AWF<br>NNF<br>STMF | 200 | 40 | 160 | 74.20% ± 2.52%<br>90.65% ± 0.71%<br>93.15% ± 2.65%<br>95.59% ± 0.07% |
| 900 rpm | OSD<br>AWF<br>NNF<br>STMF | 200 | 40 | 160 | 84.97% ± 1.67%<br>84.10% ± 1.09%<br>90.25% ± 2.30%<br>94.48% ± 0.36% |
| 1100 rpm | OSD<br>AWF<br>NNF<br>STMF | 200 | 40 | 160 | 82.57% ± 1.88%<br>93.15% ± 2.65%<br>90.25% ± 2.02%<br>94.79% ± 0.24% |
| 1300 rpm | OSD<br>AWF<br>NNF<br>STMF | 200 | 40 | 160 | 80.67% ± 1.66%<br>82.37% ± 2.90%<br>83.97% ± 3.77%<br>86.17% ± 0.30% |

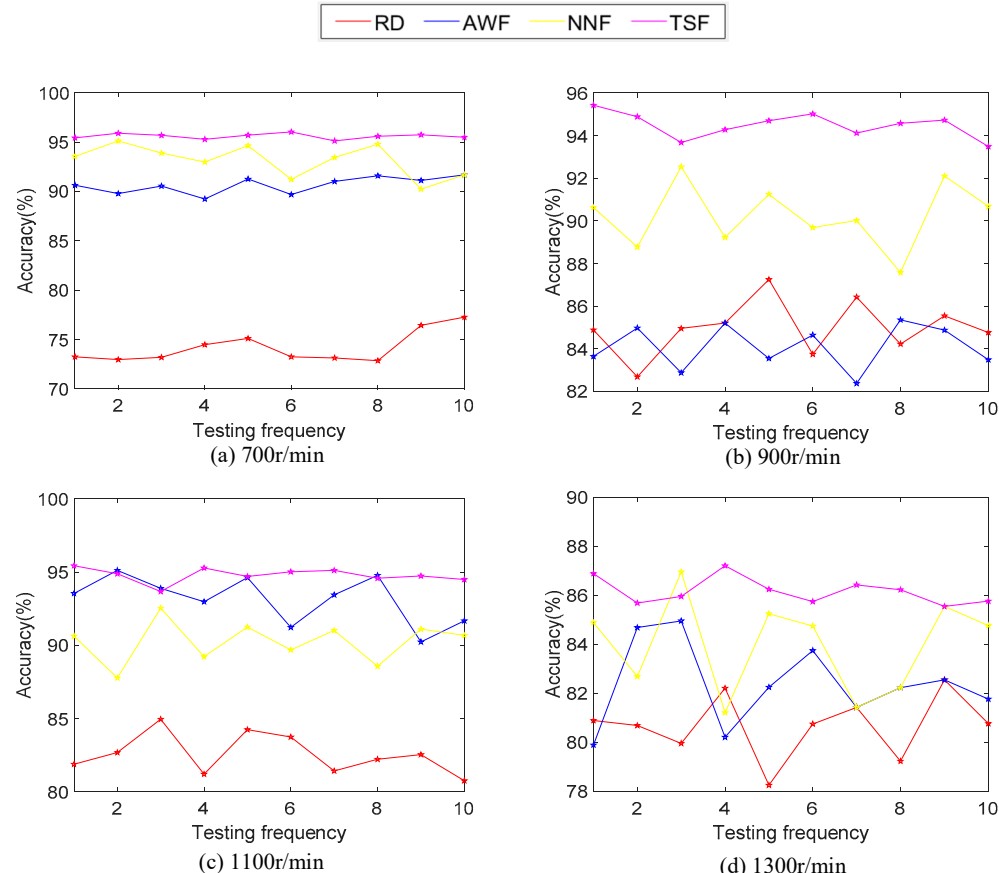

**Figure 14.** Accuracy difference diagram under different speed conditions.

In particular, the first natural frequency of the test system is 1500 rpm. In the test, when the rotation speed is 1300 rpm, the noise and vibration amplitude of the entire test bench are very high, which is close to the first natural frequency range of the test system, which results in the loss of the accuracy of the signal collection of the faulty parts. Under the working condition of 1300 rpm, the fault diagnosis rate of various methods is low, but

STMF can be compared with other signal fusion methods to highlight the superiority of the fusion method.

The STMF method has good dynamic adaptability because it is a weight distribution for the correlation analysis of real-time data. For any signal measured by any sensor, its weight will change with the support of other signals, so it has good dynamic adaptability. STMF method removes the noise from the original data through the wavelet transform method, which leads to the weight distribution of the method not beingaffected by the relevant noise, but only related to the support degree of other signals obtained by the measured useful signal. Therefore, the proposed method has good anti-interference performance.

In Figure 14, the diagnostic accuracy of STMF data is the most stable under different speed conditions, and the overall accuracy of OSD is low. The fault diagnosis accuracy and volatility of AWF and NNF will vary with speed conditions. In short, the STMF method has good adaptability and accuracy.

## 5. Conclusions

Multiple sensors are used to obtain fault information from different angles, which greatly makes up for the deficiency of single sensor information. The information fusion of different sensors and different angles is complementary to each other, which provides a good data basis for fault diagnosis.

A multi-correlation spatiotemporal fusion method is proposed by linearizing the original data of each sensor. The spatial position correlation and time synchronization of different sensors are established, and the adaptive weights are obtained by combining the signal energy correlation. The output fusion data greatly improve the accuracy of fault diagnosis.

Compared with the OSD method, AWF method and NNF method, the STMF method has great advantages in fault diagnosis accuracy and accuracy fluctuation under different velocity conditions, and the STMF method can reach a diagnosis accuracy of more than 94% at 700 rpm, 900 rpm and 1100 rpm, which provides an effective method for the development of fault diagnosis.

**Author Contributions:** Conceptualization, L.C. and S.L.; methodology L.C. and S.L.; software, L.C.; validation, L.C. and R.D.; formal analysis, L.C. and J.L.; investigation S.L.; resources S.L.; data curation K.X.; writing—original draft preparation L.C.; writing—review and editing L.C.; visualization L.C.; supervision R.D.; project administration S.L. and X.L.; funding acquisition L.C., J.L. and S.L. All authors have read and agreed to the published version of the manuscript.

**Funding:** This research was funded by the National Key Research and Development Program of China (2018YFB2003300), the National Natural Science Foundation of China (51975276) and the Major National Science and Technology Projects (2017-IV-0008-0045).

**Institutional Review Board Statement:** There are no relevant statements for this research.

**Informed Consent Statement:** Not applicable.

**Data Availability Statement:** No new data was created or analyzed in this study. Data sharing does not apply to this article.

**Acknowledgments:** Thanks to my mentor and team for this research, which was funded by the National Key Research and Development Program of China (2018YFB2003300), the National Natural Science Foundation of China (51975276), and the Major National Science and Technology Projects (2017-IV-0008-0045).

**Conflicts of Interest:** The authors declare no conflict of interest.

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
