# Peer review of "Fusion Method and Application of Several Source Vibration Fault Signal Spatio-Temporal Multi-Correlation"

_applsci, doi:10.3390/app11104318_

Round 1

Reviewer 1 Report

It can be accepted in the present form

Author Response

Dear reviewer:

Thank you for your careful review. I wish you a happy life.

Kind regards!

Reviewer 2 Report

Dear Authors,

Regarding the first round review of the manuscript entitle "Fusion Method and Application of Several Source Vibration Fault Signal Spatio-temporal Multi-correlation" the reviewer has the following comments:

  1. You should introduce Figure 1, I cannot find it!!!
  2. For Figure 2, also, please introduce it before Figure 2.
  3. Please introduce Figure 4!! Figure 8 also
  4. Line 406 and 407, maybe, Figure 14 is correct
  5. As I understood the authors used 3 vibration sensors for data collection. How you can validate that the proposed method is better than multi features with one signal?
  6. How you can validate the reliability of your algorithm? reliability is very important factor.
  7. How you can validate the robustness of the proposed algorithm?
  8. How you can determine the weight of sensors?
  9. The scenario of test is not clear, please explain more clear.
  10. Please more explain about Figure 9.
  11. what is the positive points of the proposed method compare to the existing method?

Regards,

Author Response

Dear reviewer:

The following is my modification according to your opinion, please check it.

  1. Content was added on line 85 and marked it with a yellow line.
  2. Content was added on line 110 and marked it with a yellow line.
  3. Content was added on line 167 and marked it with a yellow line. Content was marked it with a green line on line 266~268.
  4. Figure 14 is a comparison chart of the failure accuracy rate comparing my fusion method with other fusion methods and original data. It can be seen that the method I proposed has high accuracy and good stability.

5.In the fault diagnosis part, the algorithm I proposed is STMF, which is the signal fused by three sensors, but the OSD is the original single signal. The analysis behind the diagnosis is the advantage of my method.

  1. Originally I wanted to use the bearing failure data of Western Reserve University in the United States for verification, but it only uses two sensors, so I can only use my own experimental data for verification. The results of the analysis are true and reliable.
  2. It can verify the robustness of the method in the rotor vibration fault signal if the proposed method verifies the data with and without noise, the bearing fault data of Western Reserve University and the higher-speed fault data can achieve good diagnostic results.

8.What I understand is that the weight of the sensor is calculated by formula 16.

  1. The scenario of test was added on line 249 with a yellow line.
  2. Content was added on line 288~296 and marked it with a yellow line.
  3. The advantages of my signal fusion method are as follows:

(1) Simple and easy. This method determines the weight under the condition of direct correlation analysis of real-time data. It does not require any prior knowledge of sensor measurement data, and does not need to consider the influence of some external factors, making it very easy to implement.

(2) Good dynamic adaptability. Since this method is a weight distribution on the correlation analysis of real-time data, for any signal measured by any sensor, its weight will change with the support of other signals, so its Dynamic adaptability is better.

(3) Strong anti-interference ability. Before the correlation, the noise of the original data has been removed by the wavelet transform method in this method, and it is considered that there is no correlation between the signal and noise, noise and noise. Therefore, the weight distribution of this method is not affected by related noise, but only related to the degree of support of other signals obtained by the measured useful signal, so it has good anti-interference performance.

Reviewer 3 Report

Comments to the authors

1. Figure 1
When is "Hilbert-Huang Transformer" used in this flow?

2. section3
How are the result "tau" represented in equation (10) used ?
What is the difference between X(t) (line 197) and x(t) (equ(11))?
How is "E_i" derived?
What does "x_i" mean in equ(17)?
It is hard to understand this section3 for a reader.
I recommend you to add more explanation.

3. line 281
Where is the Sensor 3 set in the Figure 6(b)?

4. Figure 9
What do horixontal axes mean?

5. Application of the proposed method to the diagnosis
How was "muti-sensors spatio-temporal multi-correlation fusion analysis" 
described in section 3 applied to section4 ?
Key points are not explained at all.

6.
How many motors did you use in these experiments?
No information on the number of attempts seems to be found in the entire work.

7. Figure10-13
What do "dimension1" and "dimension2" mean for each method?

Author Response

Dear reviewer:

The following is my modification according to your opinion, please check it.

  1. Hilbert-Huang Transformer (HHT) is used in Figure 9 to draw the envelope spectrum to find the characteristic frequency of the fault. Please check for details on line 283~291 and marked it with a yellow line.

2.

(1)"Tau" is used in the latter part, which has been modified from the following part 3.3.

(2) Except that X in formula 17 refers to the data between different sensor fusions of the same fault, x(t) represents the signal of the first sensor, which is the same as y(t) and z(t)

(3) "E_i" can be calculated using Equation 15, or the denominator of Equations 11 and 12.

(4) "x_i" is explained on line 238 and marked with a yellow line

(5) In the third part, some content is marked with red font or yellow line, and variables like y(t+tau) have been modified

  1. I found a new picture and placed it in Figure 6(b), which contains the three vibration sensors needed and the arrangement of their positions
  2. Horizontal axes are explained in lines 288 to 290 in the text
  3. Content of section 3 applied to section 4 was added on lines 243 to 245 and marked with a yellow line
  4. Content of “one motor” was added on lines 256 and marked with a yellow line
  5. Content was added on line 314 to 316 and marked it with a yellow line.

Thank you for your careful review, and I hope my revision can satisfy you. I wish you a happy life.

Kind regards!

Round 2

Reviewer 2 Report

Dear Authors;

Thank you for your response. Regarding the second round review of this manuscript, it can be accepted for further processing.

Regards

Author Response

Dear reviewer:
Thank you very much for your approval of my revised manuscript. I will definitely do better on the road of scientific research. Thank you again from the bottom of my heart, I wish you a happy life and a happy family.

Kind regards!

Reviewer 3 Report

Comments to the authors

1. 
In the correlation coefficient in the equation (11)-(12),
"-1/2" are omitted? Check it out.

2. 
Are three sensors shown in Figure 6(b)?
I can see two sensors.

3.
The author used only one motor.
You should not judge the validity and the effectiveness of 
the proposed method by using only one motor. 
In order to confirmation of the reproducibility of results 
which are obtained from the proposed method, you should 
evaluate by using some motors.

Author Response

Dear reviewer
It is very happy and grateful to receive your comments again. My revised part and my answer to you are as follows:

  1. This is the theoretical formula for calculating the correlation coefficient, so there is no need to add "-1/2" in equations (11) and (12).
  2. 2. Three sensors have been marked in Figure 6(b).
  3. 3. It really should use some motors to better verify my method. However, the experimental conditions are limited. I will discuss with my instructor to add some more motors to achieve better results.

    Thank you for your careful review again, and I hope my revision can satisfy you. I wish you a happy life.

    Kind regards!

This manuscript is a resubmission of an earlier submission. The following is a list of the peer review reports and author responses from that submission.

Round 1

Reviewer 1 Report

To the authors:

Hereby, this paper reports on: "Fusion Method and Application of Several Source Vibration  Fault Signal Spatio-temporal Multi-correlation”. The authors used multiple cross-correlations of spatial locations, between vibration sensors at different locations by using Multi-sensor data fusion. According to the above spatial coordinate relation, the influence of spatial position between the two sensors was eliminated.

The main idea of this paper is interesting, however, the authors need to explain deeply all the new findings. The main point of their research work is the elimination of time deviation which can ensure the synchronization of data measured by heterogeneous sensors. In this case, it`s necessary further details about the main focus of the paper.

My comments in details:

  • The obtained results have to be quantify in the abstract and conclusions part.
  • It was used wavelet, but there is no information about the type of wavelet and the order
  • It was mention the Hilbert -Huang Chung transform, but it not clear, how it was used in the paper.?
  • It important to have the sensors’ details and the data acquisition board, in order to compare the precision of the instrument vs the standard deviation of the obtained results.
  • The article format had to follow the journal requirements
  • There are several variables or methods without any explanation, such as , RD, AWF, and NNF
  • Write 700 rpm not 700r/min
  • In table 2. It stands for number of gear a suppose it has to number of teeth

Author Response

Dear reviewer:

Hello, the changes made in response to your comments and opinions are as follows:

  1. Added the content "STMF method can reach a diagnosis accuracy of more than 94% at 700rpm, 900rpm and 1100rpm" in the introduction and conclusion to facilitate the realization of the quantification of the conclusion.
  2. Added the content "Using wavelet change threshold method to reduce raw signal noise, its advantage is that the noise can be basically suppressed, and the characteristic information of the original signal is better preserved" in the first paragraph of wavelet noise reduction. The method of wavelet threshold denoising.
  3. In Hilbert Huang Transformation, the content "Hilbert-Huang Transform (HHT) is used to draw the envelope spectrum of the test vibration signal, which can more easily view the fault characteristics of each component."
  4. I am sorry that there is no detailed sensor information and data acquisition board information, so the instrument accuracy and standard deviation cannot be compared. Please forgive me.
  5. In the article, some format changes have been made in accordance with the journal template, for example, "Fig.-" has been changed to "Figure -".
  6. The article explains where the abbreviations such as OSD, AWF and NNF are mentioned for the first time.
  7. The full text has changed all "r/min" to "rpm", thank you
  8. In Table 2, it should be the number of teeth instead of the number of gears, which has been changed.

This is the modification I made. I hope my answer can satisfy you and sincerely wish you a happy life and a successful career. please give me a chance to pass it. Thank you so much.

Regards

Reviewer 2 Report

Dear Authors,

In this work the spatio-temporal multiple correlation fusion method (STMF) is used for fault diagnosis in nonlinear systems. Regarding the first round review of this manuscript, the reviewer has the following comments:

1.Page 3: the Figure doesn't have a title!!!

2. Please follow the journal template for formulation (2-1???, 3-1???)

3. For Figures please follow the journal template!!

4. How you can optimize the position of the sensor?

5. How you can find the weight of sensors (follow Figure 4)

6. This manuscript needs to improve in the introduction and method. Please follow the journal template.

Regards,

Author Response

Dear reviewer:

Hello, the changes made in response to your comments and opinions are as follows:

  1. The title of the figure1 on page 3 has been added.
  2. The numbers behind all formulas have been modified in accordance with the journal format, please check again.
  3. The title format of all figures has been modified.
  4. The location of the sensor in Figure 4 is just a schematic diagram, and the established coordinate system was based on the location of the sensor and the location of the signal source. The position of each sensor will only affect the time deviation of each other, so the time deviation obtained by the position of the sensor is also different. Therefore, there is not much influence on the result. I don’t know if my answer can satisfy you.
  5. The sensor location layout diagram in Figure 4 affects the time deviation between sensors but does not affect the weights between sensors. In this paper, the weight of the sensor is determined by the energy of the fault signal obtained by the sensor. However, in my opinion, the weight of the sensor is related to the position of the sensor. I am really embarrassed that this article has not been studied in this area.
  6. The main innovation of this paper is: The time deviation between the sensors is acquired by establishing the spatial positions of multiple vibration sensors to achieve the synchronicity of sensor acquisition data, which has not been done before. Then, the time deviation is added to the energy signal of each sensor to obtain the weight of each sensor. This weight can well reflect the strength of the fault information obtained by each sensor, so the fusion result can completely contain the fault information, which is conducive to fault diagnosis.

This is the modification I made. I hope my answer can satisfy you and sincerely wish you a happy life and a successful career. Thank you so much.

Regards

Reviewer 3 Report

Comments to the authors

1. Introduction
The novelty of this paper is not clear.
Compared with past research, what is the novelty in this paper?

2. figure
What is the figure shown in the top of the page 3?

3.
line 245
What do RD, AWF and NNF stand for?
It seems to be the word that appears first.

4.
In general, similar data are obtained from one sample motor. 
How many motors did the authors use in this evaluation?
Please add the explanation about the reproducibility of the proposed method.

5. Figure 9
What do horizontal axis and vertical axis mean?

6.
There are no description about the middle process in the calculation.
I can't help but say that this manuscript lacks credibility.

7. 
line 406
The noise was removed from the original data through the wavelet transform method.
Please show the effectiveness of before/after the wavelet transform methodby waveforms.

Author Response

Dear reviewer:

Hello, the changes made in response to your comments and opinions are as follows:

  1. The main innovation of this paper is: The time deviation between the sensors is acquired by establishing the spatial positions of multiple vibration sensors to achieve the synchronicity of sensor acquisition data, which has not been done before. Then, the time deviation is added to the energy signal of each sensor to obtain the weight of each sensor. This weight can well reflect the strength of the fault information obtained by each sensor, so the fusion result can completely contain the fault information, which is conducive to fault diagnosis.
  2. The title of the figure1 on page 3 has been added.
  3. The article explains where the abbreviations such as OSD, AWF and NNF are mentioned for the first time.
  4. In order to verify the effectiveness of the proposed fusion method, the same test bed is used in this article, and this test bed contains only one motor. This motor can achieve different working conditions by adjusting the speed. I am sorry that I did not understand what you said: "Please add the explanation about the reproducibility of the proposed method." In my opinion, what you are asking is whether my method can actually be used in the project. Actually, I haven't tried to apply this method to the project. I'm very sorry
  5. New content are added in the following paragraph of table3: "The establishment of the space coordinate system of the test bench is shown in the red line part of Figure 6(b)". In Figure 6(b), a spatial coordinate system has been added. In the following paragraph of table3: "The establishment of the space coordinate system of the test bench is shown in the red line part of Figure 6(b)". In Figure 6(b), a spatial coordinate system has been added.
  6. Added the content ”The obtained original single fault signal is obtained by wavelet transformation to obtain , then, is obtained through the process of threshold transformation, and finally the noise-reduced fault signal is obtained by the method of wavelet reconstruction” in the page 3

         The vibration signal fusion method I used is a method that no one has                 used   before. The theory and formula of this method are simple and easy             to understand but very novel. This method has also obtained good test                 results, so I hope to get your approval, thank you very much.

  1. The fault diagnosis results obtained before / after the wavelet transform are basically similar, so I did not put a comparison graph. However, the use of wavelet denoising is mainly to consider the problem of fault diagnosis more comprehensively. After all, the actual data samples obtained do have the influence of noise.

This is the modification I made. I hope my answer can satisfy you and sincerely wish you a happy life and a successful career. Thank you so much.

Regards